# Invariant Representations without Adversarial Training

**Daniel Moyer, Shuyang Gao, Rob Brekelmans, Greg Ver Steeg, and Aram Galstyan**
Information Sciences Institute
University of Southern California
{moyerd, gaos, brekelma}@usc.edu   {gregv, galstyan}@isi.edu

## Abstract

Representations of data that are invariant to changes in specified factors are useful for a wide range of problems: removing potential biases in prediction problems, controlling the effects of covariates, and disentangling meaningful factors of variation. Unfortunately, learning representations that exhibit invariance to arbitrary nuisance factors yet remain useful for other tasks is challenging. Existing approaches cast the trade-off between task performance and invariance in an adversarial way, using an iterative minimax optimization. We show that adversarial training is unnecessary and sometimes counter-productive; we instead cast invariant representation learning as a single information-theoretic objective that can be directly optimized. We demonstrate that this approach matches or exceeds performance of state-of-the-art adversarial approaches for learning fair representations and for generative modeling with controllable transformations.

## 1   Introduction

The removal of unwanted information is a surprisingly common task. Transform-invariant features in computer vision, "fair" encodings from the algorithmic fairness community, and two-stage regressions often used in scientific studies are all cases of the same general concept: we wish to remove the effect of some outside variable $c$ on our data $x$ while still relevant to our original task. In the context of representation learning, we wish to map $x$ into an encoding $z$ that is uninformative of $c$, yet also optimal for our task loss $\mathcal{L}(z, \dots)$.

These objectives are often operationalized as an independence constraint $z \perp c$. Encodings satisfying this condition are invariant under changes in $c$, thus called "invariant representations". In practice these constraints are often relaxed to other measures; in recent works an adversary's ability to predict $z$ from $c$ has been used as such a proxy [15, 21], transplanting adversarial losses from generative literature to encoder/decoder settings.

In the present work we instead relax $z \perp c$ to a penalty on the mutual information $I(z, c)$. We provide an analysis of this loss, showing that:

1. $I(z, c)$ admits a useful variational upper bound. This is in contrast to the usual lower bound e.g. bounds on $I(z, y)$ for some labels $y$.

2. When placed alongside the Variational Auto Encoder (VAE) and the Variational Information Bottleneck (VIB) frameworks, the upper bound on $I(z, c)$ produces a computationally tractable form for learning $c$-agnostic encodings and predictors.

3. The adversarial approach can be also derived as a procedure to minimize $I(z, c)$, but does not provide an upper bound.

Our proposed methods have the practical advantage of only requiring $c$ at training time, and not at test time. They are thus viable for production settings where accessing $c$ is expensive (requiring human labeling), impossible (requiring underlying transformations), or legally inadvisable (sharing protected data). On the other hand, our method also produces a conditional decoder taking both $c$ and $z$ as inputs. While for some purposes this might be discarded at test time, it also can be manipulated to function similarly to Fader Networks [14], in that we can generate realistic looking transformations of an input image where some class label has been altered. We empirically test our proposed $c$-agnostic VAE and VIB methods on two standard "fair prediction" tasks, as well as an unsupervised learning task demonstrating Fader-like capabilities.

## 1.1   Related Work

The removal of covariate factors from scientific data has a long history. Observational studies in the sciences often cannot control for every factor influencing subjects, thus a large amount of literature has been generated on the topic of removing such factors after data collection. Simple statistical techniques usually involve modifying analyses with corresponding covariate effects [18] or sometimes multi-level regressions [7]. Modern methods have included more nuanced feature generation and more complex models, but follow along the same vein [5, 6] "Regressing out" or "controlling for" unwanted factors can be effective, but place strong constraints on later analyses (namely, the observation of the same unwanted covariates).

A similar concept also has deep roots in computer vision, where transform-invariant features and/or methods have been sought after for some time. Often these methods were designed for specific cases, e.g. scale-invariant or rotation invariant features. Early examples include Steerable Filters [8, 11], and later SIFT [16]. For many image transformations, data augmentation has become a standard practical tool to encourage invariance.

Recent work has provided a group-theoretic [4] analysis of the removal of covariate information (either by design or by augmentation), in which equivalences are drawn between finding invariant features and finding the quotient space of the domain over a covariate group action. More generally, an empirical solution was proposed by Lample et al. [14], who removed specific visual features from a latent representation through adversarial training.

More recently the algorithmic fairness community has investigated fair methods [12] and fair representations [22]. Derived in part from the desire to avoid discriminating against protected classes of individuals (and in part to avoid breaking laws and/or to avoid being the subject of a civil suit), the objective of these methods has been to preserve task accuracy (usually classification or regression) while removing bias against the protected class of individuals.

Particularly relevant to our work are the recent methods proposed by Louizos et al. [15] and Xie et al. [21], which have a similar problem setup. Both methods generate representations that make it difficult for an adversary to recover the protected class but are still useful for a classification task. Louizos et al. propose the "Variational Fair Auto-Encoder" (VFAE), which, as its name suggests, modifies the VAE of Kingma and Welling [13] to produce fair[1] encodings, as well as a supervised case providing fair classifications. Xie et al. combine this concept with adversarial training, adding (inverted) gradient information to produce fair representations and classifications. This adversarial solution coincides exactly with the conceptual framework used in a computer vision application for constructing Fader Networks [14].

Compressive encoding in a learning context is also well studied. In particular, the Information Bottleneck [20] and its modern successor Variational Information Bottleneck (VIB) [3, 2] both provide compressive encodings, aiming for "relevance" with respect to a target variable (usually a label). An unsupervised method, CorEx, also has a similar extension (Anchored Corex [9]), in which latent factors can be driven toward specific targets. Our work could be thought of as adding "negative anchors" or aiming for "irrelevance" with respect to protected classes.

Models including "nuisance" factors were also considered by Soatto and Chiuso [19], in which the authors propose definitions of both nuisances and invariant representations. The authors follow the group theoretic concept of nuisances. Achille and Soatto [1] directly utilize these results, providing

the same criterion and relaxation for invariance we will use here, minimal mutual information between the representation $z$ and the covariate $c$. While nuisances form only a small subsection of the paper, the authors propose and test a sampling based approach to learning invariant representation. Their method is predicated on the ability to sample from the nuisance distribution (e.g. adding occlusions in images). We optimize a similar objective, but avoid such practical constraints.

## 2  Model

Consider a general task that includes an encoding of observed data $x$ into latent variables $z$ through the conditional likelihood $q(z|x)$ (an encoder). Further assume that we observe a variable $c$ which exhibits statistical dependence with $x$ (possibly non-linear dependence). We would like to find a $q(z|x)$ that minimizes our loss function $\mathcal{L}$ from the original task, but that also produces a $z$ independent of $c$.

This is clearly a difficult optimization; independence is a very strong condition. A natural relaxation of this is the minimization of the mutual information $I(z, c)$. We can write our relaxed objective as

$$\min_q \mathcal{L}(q, x) + \lambda I(z, c) \tag{1}$$

where $\lambda$ is a trade-off parameter between the two objectives. $\mathcal{L}$ might involve other variables as well, e.g. labels $y$. Without details of $\mathcal{L}$ and its associated task, we can still provide insight into $I(z, c)$.

Before continuing it is important to note that all entropic quantities related to $z$ are from the encoding distribution $q$ unless explicitly stated otherwise. In some cases entropies depend on prior distributions, $p(z)$, and this will be explicitly noted.

From properties of mutual information, we have that $I(z, c) = I(z, x) - I(z, x|c) + I(z, c|x)$. Here, we note that $q(z|x)$ is the function that we are optimizing over, and thus the distribution of $z$ solely depends on $x$. Thus,

$$I(z, c|x) = H(z|x) - H(z|x, c) = H(z|x) - H(z|x) = 0. \tag{2}$$

Using Mutual Information properties and a variational inequality, we can then write the following:

$$I(z, c) = I(z, x) - I(z, x|c) \tag{3}$$

$$= I(z, x) - H(x|c) + H(x|z, c) \tag{4}$$

$$\leq I(z, x) - H(x|c) - \mathbb{E}_{x, c, z \sim q}[\log p(x|z, c)] \tag{5}$$

$$= \mathbb{E}_{z, x}[\log q(z|x) - \log q(z)] - H(x|c) - \mathbb{E}_{x, c, z \sim q}[\log p(x|z, c)] \tag{6}$$

$$= \mathbb{E}_x[\, KL[\, q(z|x) \,\|\, q(z) \,]\,] - H(x|c) - \mathbb{E}_{x, c, z \sim q}[\log p(x|z, c)]. \tag{7}$$

$H(x|c)$ is a constant and can be ignored. In Eq. 5 we introduce the variational distribution $p(x|z, c)$ which will play the traditional role of the decoder. $I(z, c)$ is thus bounded up to a constant by a divergence and a reconstruction error.

The result is similar in appearance to the bound from Variational Auto-Encoders [13], wherein we balance the divergence between $q(z|x)$ and a prior $p(z)$ against the reconstruction error. Here our penalty on $I(z, c)$ amounts to encouraging $q(z|x)$ to be close to its marginal $q(z)$, i.e. to vary less across inputs $x$, no matter the form of $q(z|x)$ or $q(z)$. From a coding viewpoint our penalty encourages the compression of $x$ out of $z$ using the $I(z, x)$ term from Eq 5.

In both interpretations, these penalties are tempered by conditional reconstruction error. This provides additional intuition; by adding a copy of $c$ into the reconstruction, we ensure that compressing away information in $z$ about $c$ is *not* penalized. In other words, conditional reconstruction combined with compressing regularization leads to invariance w.r.t. to the conditional input.

### 2.1  Invariant Codes through VAE

We apply our proposed penalty to the VAE of Kingma and Welling [13], inspired by the similarity of the penalty in Eq. 7 to the VAE loss function. The original VAE stems from the classical unsupervised task of constructing latent factors, $z$, so that $p(z), p(x|z)$ define a generative model that maximizes the log likelihood of the data $\mathbb{E}_x[\log p(x)]$. This generally intractable expression is lower bounded using Jensen's inequality and a variational approximation:

$$\log p(x) \geq -KL[\, q(z|x) \,\|\, p(z) \,] + \mathbb{E}_{z \sim q(z|x)}[\log p(x|z)]. \tag{8}$$

Kingma and Welling [13] frame $q(x|z)$ and $p(z|x)$ as an encoder/decoder pair. They then provide a re-parameterization trick that, when used with standard function approximators (neural networks), allows for efficient estimation of latent codes $z$. In short, the reparametrization is the following:

$$q(z|x) = g_\theta(x) + \varepsilon, \qquad \varepsilon \sim \mathcal{N}(0, \sigma(\theta)) \tag{9}$$

where $g_\theta$ is a deterministic function (neural network) with parameters $\theta$, and $\varepsilon$ is an independent random variable from a Normal distribution[2] also parameterized by $\theta$.

We can reformulate Kingma and Welling's VAE to include our penalty on $I(z, c)$. Define $\{x_i\}_{i=1}^N$ data and latent factors $\{z_i\}$, but also define observed $\{c_i\}$ upon which $x$ may have non-trivial dependence. That is,

$$p(x, z, c) = p(z, c)p(x|z, c). \tag{10}$$

The invariant coding task is to find $q(z|x), p(x|z, c)$ that maximize $\mathbb{E}_{(x,c)}[\log p(x|c)]$, subject to $z \perp c$ under $z \sim q$ (i.e. subject to the estimated code $z$ being invariant to $c$). We make the same relaxation as in Eq. 1 to formulate our objective:

$$\max \mathbb{E}_{(x,c)}[\log p(x|c)] - \lambda I(z, c). \tag{11}$$

Starting with the first term, we can derive a familiar looking encoder/decoder loss function that now includes $c$.

$$\log p(x|c) = \log \int p(x, z|c)dz \tag{12}$$

$$= \log \int \frac{p(x, z|c)}{q(z|x)} q(z|x)dz = \log \mathbb{E}_{z \sim q}\left[\frac{p(x, z|c)}{q(z|x)}\right] \tag{13}$$

$$\geq \mathbb{E}_{z \sim q}[\log p(x, z|c) - \log q(z|x)] \tag{14}$$

$$= \mathbb{E}_{z \sim q}[\log p(z|c) - \log q(z|x) + \log p(x|z, c)]. \tag{15}$$

Because $p(z|c)$ is a prior, we can make the assumption that $p(z|c) = p(z)$, the prior marginal distribution over $z$. This is a willful model misspecification: for an arbitrary encoder, the latent factors, $z$, are probably not independent of $c$. However, practically we wish to find $z$ that *are* independent of $c$, thus it is reasonable to include such a prior belief in our generative model. Taking this assumption, we have

$$\log p(x|c) \geq \mathbb{E}_{z \sim q}[\log p(z) - \log q(z|x)] + \mathbb{E}_{z \sim q}[\log p(x|z, c)] \tag{16}$$

$$= -KL[\, q(z|x) \,\|\, p(z) \,] + \mathbb{E}_{z \sim q}[\log p(x|z, c)]. \tag{17}$$

This is almost exactly the same as the VAE objective in Eq. 8, except our decoder $p(x|z, c)$ requires $c$ as well as $z$. Putting this together with the penalty term Eq. 7, we have the following variational bound on the combined objective (up to a constant):

$$\mathbb{E}_{(x,c)}[\log P(x|c)] - \lambda I(z, c) \geq$$
$$\mathbb{E}_{(x,c)}\big[ -KL[\, q(z|x) \,\|\, p(z) \,] - \lambda KL[\, q(z|x) \,\|\, q(z) \,] + (1 + \lambda)\mathbb{E}_{z \sim q}[\log p(x|z, c)] \,\big]. \tag{18}$$

We use this bound to learn $c$-invariant auto-encoders.

### 2.1.1 Derivation of an approximation for the Conditional-Marginal divergence

Equation 18 is our desired loss function for learning invariant codes $z$ in an unsupervised context. Unfortunately it contains $q(z)$, the empirical marginal distribution of latent code $z$, which is difficult to compute. Using the re-parameterization trick, $q(z)$ becomes a mixture distribution and this allows

us to approximate its divergence from $q(z|x)$.

$$KL[q(z|x)\|q(z)] = -H(q(z|x)) - \int q(z|x) \log \int q(z|x')p(x')dx'dz \tag{19}$$

$$\approx -H(q(z|x)) - \int q(z|x) \log \frac{1}{B} \sum_{x'} q(z|x')dz \tag{20}$$

$$= -H(q(z|x)) - \mathbb{E}_{z\sim q|x}[\log \sum_{x'} q(z|x')] - \log B \tag{21}$$

$$\leq -H(q(z|x)) - \sum_{x'} \mathbb{E}_{z\sim q|x}[\log q(z|x')] - \log B \tag{22}$$

$$= \sum_{x'}[-H(q(z|x)) - \mathbb{E}_{z\sim q|x}[\log q(z|x')]] + (B-1)H(q(z|x)) - \log B \tag{23}$$

$$= \sum_{x'}\underbrace{KL[q(z|x)\|q(z|x')]}_{\text{KL between Gaussians}} + (B-1)\underbrace{H(q(z|x))}_{\text{Gaussian Ent.}} - \log B \tag{24}$$

We can thus approximate $KL[q(z|x)\|q(z)]$ from its pairwise distances $KL[q(z|x)\|q(z|x')]$, which all have a closed form due to the reparameterization trick. While this requires $O(b^2)$ operations for batch size $b$, we can reduce pairwise Gaussian KL divergence to matrix algebra, making this computation fast in practice.

This further provides insight into the previously proposed Variational Fair Auto-Encoder of Louizos et al [15]. In that paper, the authors add a Maximum Mean Discrepancy penalty as a somewhat ad hoc regularizer. This nevertheless works in practice quite well, as it encourages the statistical moments of each $q(z|c)$ to be the same over the varying values of $c$. Our condition of $KL[q(z|x)\|q(z)]$ has equivalent minima, and shares the "q-regularizing" flavor of the MMD penalty.

### 2.1.2 Alternate derivation leads to adversarial loss

In Equation 3 we used the identity $I(z,c) = I(z,x) - I(z,x|c) + I(z,c|x)$, with the caveat that the third term $I(z,c|x)$ is zero. We could have instead used another identity, $I(z,c) = H(c) - H(c|z)$. Here, the first term is constant, but expanding the second term provides the following:

$$H(c|z) = \mathbb{E}_{c,z\sim q}[-\log p(c|z)] \tag{25}$$

$$= \inf_{r(c|z)} \mathbb{E}_{c,z\sim q}[-\log r(c|z)] \tag{26}$$

$$\mathbb{E}_{(x,c)}[\log P(x|c)] - \lambda I(z,c) \geq$$
$$\mathbb{E}_{(x,c)}[-KL[q(z|x)\|p(z)] + \mathbb{E}_{z\sim q}[\log p(x|z,c)]] + \lambda \inf_{r(c|z)} \mathbb{E}_{c,z\sim q}[-\log r(c|z)] \tag{27}$$

The last inequality is again up to a constant term. Interpreting this in machine learning parlance, another possible approach for minimizing $I(z,c)$ is to optimize the conditional distribution $p(c|z)$ so that $r$, the lowest entropy predictor of $c$ given $z$, has the highest entropy (i.e. is as inaccurate as possible at predicting $c$). This is often operationalized by adversarial learning, and subsequent error may be due in part to the adversary not achieving the infimum. Practically speaking, this may indicate that over-training adversaries would benefit performance by bringing the adversarial gradient closer to the infimum adversary's gradient.

### 2.2 Supervised Invariant Codes through the Variational Information Bottleneck

Learned encodings are often used for downstream supervised prediction tasks. Just as in Variational Fair Auto-encoders [15], we can model both at the same time to offer $c$-invariant predictions. Our formulation of this problem fits into the Information Bottleneck framework [20] and mirrors the Variational Information Bottleneck (VIB) [3].

Conceptually, VAEs have strong connections to the Information Bottleneck [3]. Stepping out of the generative context, we can "reroute" our decoder to a label variable $y$. This gives us the following computational model:

$$x \xrightarrow{q(z|x)} z \xrightarrow{p(y|z)} y \tag{28}$$

The bottleneck paradigm prescribes optimizing over $q$ and $p$ so that $I(z, y)$ is maximal while minimizing $I(x, z)$ ("maintaining information about y with maximal compression of x into z"). As illustrated by Alemi et al. [3], this can be approximated using variational inference.

We can produce $c$-invariant codes in the supervised Information Bottleneck context using the relaxation from Eq. 1. Beginning with the bottleneck objective $\max_{q,p} I(z, y) - \beta I(x, z)$ and then including the minimization of $I(z, c)$, we have

$$\max_{p,q} I(z, y) - \beta I(x, z) - \lambda I(z, c) \tag{29}$$

We can then apply the same bound as in Eq. 5 to obtain, up to constant $H(x|c)$, the following:

$$\max_{p,q} I(z, y) - (\beta + \lambda) I(x, z) + \lambda \mathbb{E}[\log p(x|z, c)]. \tag{30}$$

In this objective we have a maximization of likelihood $p(x|z, c)$. This is a decoder loss, adding a third branch to our network. Following the derivation in Alemi et al. [3] as well as a similar path as in Section 2.1, the variational bound on the objective is

$$\begin{aligned} I(z, y) - (\beta &+ \lambda) I(x, z) + \lambda \mathbb{E}[\log p(x|z, c)] \geq \\ &\mathbb{E}_{(x,c)}[\, \mathbb{E}_{z,y}[\log p(y|z)] - (\beta + \lambda) KL[\, q(z|x) \parallel q(z) \,] + \lambda \mathbb{E}_z[\log p(x|z, c)] \,]. \end{aligned} \tag{31}$$

We use Eq. 31 to learn $c$-invariant predictors. Optimization is performed over three function approximations: one encoder $q(z|x)$, one conditional decoder $p(x|z, c)$, and one predictor $p(y|z)$. We further must compute $KL[\, q(z|x) \parallel q(z) \,]$ from the $I(x, z)$ penalty term. Instead of following Alemi et al.[3], we again use the approximation to $KL[\, q(z|x) \parallel q(z) \,]$ from Eq. 24.

## 3 Computation and Empirical Evaluation

We have two modified VAE loss (Eq. 18) and modified VIB loss (Eq. 31). In both we have to learn an encoder and decoder pair $q(z|x)$ and $p(x|z, c)$. We use feed forward networks to approximate these functions. For $q(z|x)$ we use the Gaussian reparameterization trick, and for $p(x|z, c)$ we simply concatenate $c$ onto $z$ as extra input features to be decoded. In the modified VIB we also have a predictor branch $p(y|z)$, which we also use a feed forward network to parametrize. Specific architectures (e.g. number of layers and nodes per layer for each branch) vary by domain.

We evaluate the performance on of our proposed invariance penalty on two datasets with a "fair classification" task. We also demonstrate "Fader Network"-like capabilities for manipulating specified factors in generative modeling on the MNIST dataset.

### 3.1 Fair Classification

For each fair classification dataset/task we evaluated both prediction accuracy and adversarial error in predicting $c$ from the latent code. We compare against the Variational Fair Autoencoder (VFAE) [15], and the adversarial method proposed in Xie et al. [21]. Both datasets are from the UCI repository. The preprocessing for both datasets follow Zemel et al. 2013[22], which is also the source for the pre-processing in our baselines [15, 21].

The first dataset is the German dataset, containing 1000 samples of personal financial data. The objective is to predict whether a person has a good credit score, and the protected class is Age (which, as per [22], is binarized). The second dataset is the Adult dataset, containing 45,222 data points of US census data. The objective is to predict whether or not a person has over 50,000 dollars saved in the bank. The protected factor for the Adult dataset is Gender[3].

Wherever possible we use architectural constraints from previous papers. All encoders and decoders are single layer, as specified by Louizos et al. [15] (including those in the baselines), and for both datasets we use 64 hidden units in our method as in Xie et al., while for VFAE we use their described architecture. We use a latent space of 30 dimensions for each case. We train using Adam using the same hyperparameter settings as in Xie et al., and a batch size of 128. Optimization and parameter tuning is done via a held-out validation set.

| German Dataset | Adv. Loss | Pred Acc. |
|---|---|---|
| Maj. Class | 0.725 | 0.695 |
| VFAE [15] | 0.717 | 0.720 |
| Xie et al. [21] | 0.811 | 0.695 |
| Proposed | 0.698 | 0.710 |

| Adult Dataset | Adv. Loss | Pred Acc. |
|---|---|---|
| Maj. Class | 0.675 | 0.752 |
| VFAE [15] | 0.882 | 0.842 |
| Xie et al. [21] | 0.888 | 0.831 |
| Proposed | 0.675 | 0.844 |

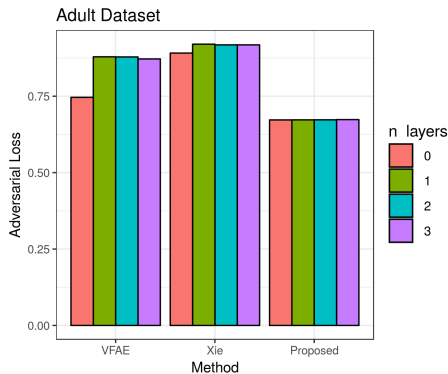

Figure 1: On the left we display the adversarial loss (the accuracy of the adversary on $c$) and predictive accurracy on $y$ for three methods, plus the majority-class baseline, on both Adult and German datasets. For **adv. loss lower is better**, while for **pred. acc. higher is better**. On the right we plot adversarial loss by varying adversarial strength (indicated by color), parameterized by the number of layers from zero (logistic regression) to three. All evaluations are performed on the hold-out test sets.

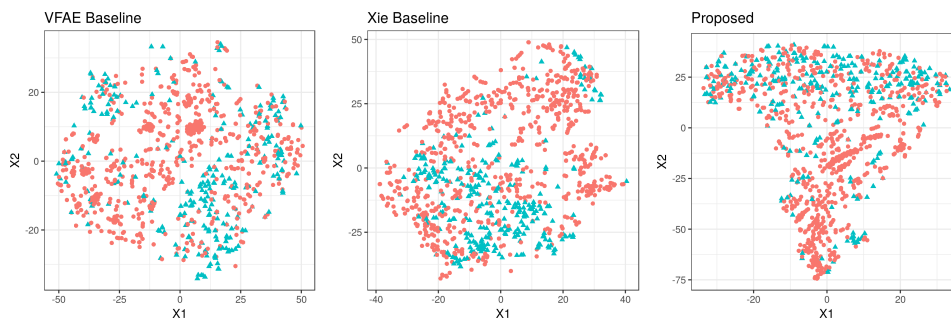

Figure 2: t-SNE plots for the latent encodings of (Left to Right) the VFAE, Xie et al., and our proposed method on the Adult dataset (first 1000 pts., test split). The value of the $c$ variable is provided as color, where red is the majority class.

For each tested method we train a discriminator to predict $c$ from generated latent codes $z$. These discriminators are trained independently from the encoder/decoder/within-method adversaries. We use the architecture from Xie et al. [21] for these post-hoc adversaries, which describes a three-layer feed-forward network trained using batch normalization and Adam (using $\gamma = 1$ and a learning rate of $0.001$), with 64 hidden units per layer, using absolute error. We generalize this to four adversaries, increasing in the number of hidden layers. Each discriminator is trained post-hoc for each model, even in cases with a discriminator in the model (e.g. the model proposed by Xie et al. [21]).

### 3.2 Unsupervised Learning

We demonstrate a form of unsupervised image manipulation inspired by Fader Networks [14] on the MNIST dataset. We use the digit label as the covariate class $c$, which pushes all non-class stylistic information into the latent space while attempting to remove information about the exact digit being written. This allows us to manipulate the decoder at test time to produce different artificial digits based on the style of one digit. We use 2 hidden layers with 512 nodes for both the encoder and the decoder.

## 4 Results

For the German dataset shown on top table of Figure 1, the methods are roughly equivalent. All methods have comparable predictive accuracy, while the VFAE and the proposed method have

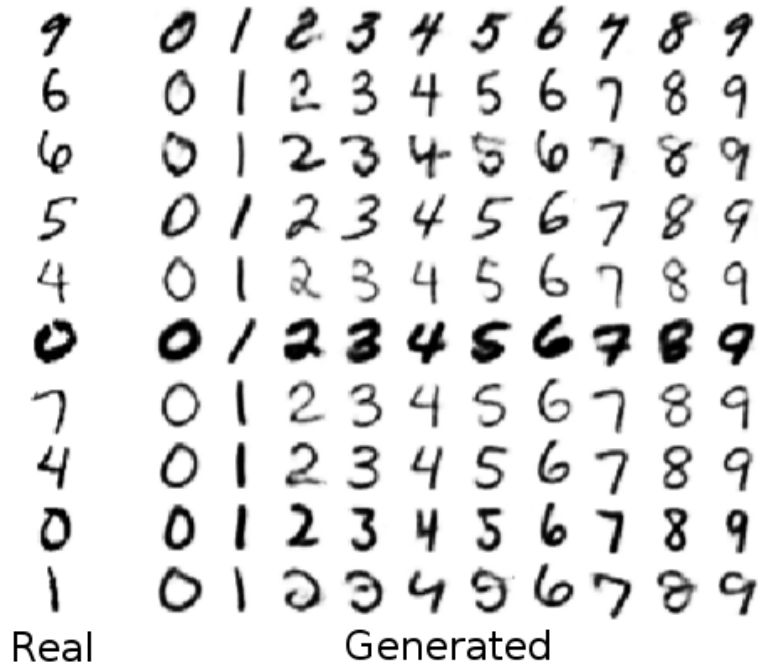

Real                    Generated

Figure 3: We demonstrate the ability to generate stylistically similar images of varying classes using the MNIST dataset. The left column is mapped into $z$ that is invariant to its digit label $c$. We then can generate an image using $z$ and any other specified digit, $c'$, as show on the right.

competitive adversarial loss. In general however, the smaller dataset does not differentiate the methods.

For the larger Adult dataset shown on the bottom table of Figure 1, all three methods again have comparable predictive accuracy. However, against stronger adversaries each baseline has very high loss. Our proposed method has comparable accuracy with the VFAE, while providing the best adversarial error across all four adversarial difficulty levels.

We further visualized a projection of the latent codes $z$ using t-SNE [17]; invariant representations should produce inseparable embeddings for each class. All methods have large red-only regions; this is somewhat expected for the majority class. However, both baseline methods have blue-only regions, while the proposed method has only a heterogenous region[4].

Figure 3 demonstrates our ability to manipulate the conditional decoder. The left column contain the actual images (randomly selected from the test set), while the right columns contain images generated using the decoder. Particularly notable are the transfer of azimuth and thickness, and the failure of some styles to transfer to some digits (usually curved to straight digits or vice versa).

## 5 Discussion

As show analytically in Section 2.1.2, in the optimal case adversarial training can perform as well as our derived method; it is also intuitively simple and allows for more nuanced tuning. However, it introduces an extra layer of complexity (indeed, a second optimization problem) into our system. In this particular case of invariant representation, our results lead us to believe that adversarial training is unnecessary.

This does not mean that adversarial training for invariant representations is strictly worse in practice. There are certainly cases where training an adversary may be easier or less restrictive than other

methods, and due to its shared literature with Generative Adversarial Networks [10], there may be training heuristics or other techniques that can improve performance.

On the otherhand, we believe that our derivations here shed light on why these methods might fail. We believe specific failure modes of adversarial training can be attributed to Eq. 27, where the adversary fails to achieve the infimum. Bad approximations (i.e. weak or poorly trained adversaries) may provide bad gradient information to the system, leading to poor performance of the encoder against a post-hoc adversary.

Our experimental results do not match those reported in Xie et al. While in general their method has comparable performance for predictive accuracy, we do not find that their adversarial error is low; instead, we find that the encoder/adversary pair becomes stuck in local minima. We also find that the adversary trained alongside the encoder performs badly against the encoder (i.e. the adversary cannot predict $c$ well), but a post-hoc trained adversary performs very well, easily predicting $c$ (as demonstrated by our experiments).

It may be that we have inadvertently built a stronger adversary. We have attempted to follow the author's experimental design as closely as possible, using the same architecture and the same adversary (using the gradient-flip trick and 3-layer feed forward networks). With the details provided we could not replicate their reported adversarial error for their method, nor for the VFAE method. However, we are able to reproduce the adversarial error reported in Louizos et al., which uses logisic regression. In general for stronger adversaries the adversarial loss will increase, but the relative rankings should remain roughly the same.

## 6    Conclusion

We have derived a variational upper bound for the mutual information between latent representations and covariate factors. Provided a dataset with labeled covariates, we can train both supervised and unsupervised learning methods that are invariant to these factors without the use of adversarial training. After training our method can be used in production without requiring covariate labels. Finally, our approach also enables manipulation of specified factors when generating realistic data. Our direct, information-theoretic optimization approach avoids the pitfalls inherent in adversarial learning for invariant representation and produces results that match or exceed capabilities of these state-of-the-art methods.

**Acknowledgements**

This work was supported by DARPA grants W911NF-16-1-0575 and FA8750-17-C-0106, as well as the NSF Graduate Research Fellowship Program Grant Number DGE-1418060. We would like to thank the conference organizers, area chairs, and especially the anonymous reviewers for their work and helpful input. We also would like to thank Ayush Jaiswal for several insightful conversations, and Ishaan Gulrajani for finding and correcting a bug in our evaluation code.

## Footnotes

[1]The definition of "fair" in an algorithmic setting is of some debate. A fair encoding in this paper is uninformative of protected classes. We offer no opinion on whether this is truly "fair" in a general or legal sense, taking the word at face value as used by Louizos et al.

[2]In the original paper, this was defined more generally; here we only consider the Normal distribution case.

[3]In some papers the protected factor for the Adult dataset is reported as Age, but those papers also reference Zemel et al. [22] as the processing and experimental scheme, which specifies Gender.

[4]Previous versions of this paper had severely contorted latent codes for the Xie et al. baseline. Further investigation showed this to be a convergence issue. Mild performance improvements were also observed.

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
