[Reviews · NeurIPS 2018]

Reviewer 1



This paper adresses the problem of learning invariant representation by defining an information-theoretic constraint defined as the mutual mutual information between the constructed representation z and the variability factor c. This constraint is further incorporated in a VAE. Mutual Information is know to be hard to compute, and in line with the very very abundant recent literature, the authors propose to use a variational approximation. I found the paper hard to follow, and I must say that I’m still struggling to write my review. I don’t find that there are original ideas in the paper, Variational Approximation of MI is now something established. Combining it in VAE seems not novel. I can change my mind on this, but I need more explanations about the real novelty. Also, I found the title misleading and unrelated to the actual purpose of the paper. Also the part on adversarial attack seems superficial to me. In the introduction, I didn’t understand why the authors discuss observational studies, and the use of the term « confounding » which refers to causal reasoning is not appropriate in this setting.

Reviewer 2



Update following rebuttal: I thank the authors for their thoughtful reply to my question. I still believe this is a good paper, but the discussion with other reviewers and a second read have convinced me that it has its limitations. I have therefore adjusted my score slightly downwards. ----- The authors present a method of penalizing mutual information between a representation and confounding variables in the context of variational autoencoders and variational information bottleneck. This corresponds to a principled, data-driven way of encouraging invariances in the representation which encode confounding variable identity. From what I can tell, the presentation is clear, connections to related work are sufficiently covered, results appear convincing. Good theoretical work! The biggest drawback of the paper in my mind, and what limits the generality/relevance of the paper, is the assumption of Gaussian encoders, which appears to be necessary to approximate the KL divergence in eq. (18), as described in section 2.1.1. Can the authors say something about how this might generalize to other choices of distributions? A small nit: equation 8 contains a sign flip.

Reviewer 3



The paper studies the timely and extremely relevant problem of learning invariant representations for data. In contrary to the main stream of works in this area, it does not consider an adversarial training approach, and this is refreshing! Instead, it proposes to solve an information-theoretic objective, which proves to be competitive and even better that state-of-the-art adversarial approaches. In more details, the authors derive a useful upper-bound on the mutual information, that can be nicely integrated in the VAE or VIB frameworks. This derivation is essentially the main contribution of the paper, and it provides an elegant and generic framework for learning invariant representations. The paper is generally well written, even it is quite dense and probably too detailed at places, like in Section 2. This sort of leads the reader to lose the main intuition behind the proposed framework. The experiments are generally well chosen, and the visualisation of Fig 2 is pretty interesting and original. The results are however not always fully convincing (see for example the German dataset in Fig 1), but it does not critically reduce the quality of the paper and the proposed ideas. Suggestions: - the title 'evading the adversary' is a bit misleading: what is done in the paper is actually learning invariant representations... - while there is an attempt to stay generic in the presentation of the framework, it sometimes leads to confusion: in particular, the mix between fairness ideas, and more classical invariant representation learning objectives is sometimes confusing (like in the related work section, and in the experiments). Overall, the paper addresses a timely problem, with a new approach, based on information theoretic concepts and bounds on the mutual information. These ideas are integrated into classical learning architectures, and leads to competitive results.